# Remote Hospital Care for Recovering COVID-19 Patients Using Telemedicine: A Randomised Controlled Trial

**DOI:** 10.3390/jcm10245940

**Published:** 2021-12-17

**Authors:** Harriët M. R. van Goor, Martine J. M. Breteler, Kim van Loon, Titus A. P. de Hond, Johannes B. Reitsma, Dorien L. M. Zwart, Cornelis J. Kalkman, Karin A. H. Kaasjager

**Affiliations:** 1Department of Internal Medicine, University Medical Centre Utrecht, 3584 Utrecht, The Netherlands; m.j.m.breteler@umcutrecht.nl (M.J.M.B.); t.a.p.dehond@umcutrecht.nl (T.A.P.d.H.); h.a.h.kaasjager@umcutrecht.nl (K.A.H.K.); 2Department of Digital Health, University Medical Centre Utrecht, 3584 Utrecht, The Netherlands; 3Department of Anaesthesiology, University Medical Centre Utrecht, 3584 Utrecht, The Netherlands; k.vanloon-2@umcutrecht.nl (K.v.L.); c.j.kalkman@umcutrecht.nl (C.J.K.); 4Julius Centre for Health Sciences and Primary Care, Department of Epidemiology, University Medical Centre Utrecht, 3584 Utrecht, The Netherlands; j.b.reitsma-2@umcutrecht.nl; 5Julius Centre for Health Sciences and Primary Care, Department of General Practice, University Medical Centre Utrecht, 3584 Utrecht, The Netherlands; d.zwart@umcutrecht.nl

**Keywords:** COVID-19, remote hospital care, remote monitoring, telemedicine

## Abstract

Background: To ensure availability of hospital beds and improve COVID-19 patients’ well-being during the ongoing pandemic, hospital care could be offered at home. Retrospective studies show promising results of deploying remote hospital care to reduce the number of days spent in the hospital, but the beneficial effect has yet to be established. Methods: We conducted a single centre, randomised trial from January to June 2021, including hospitalised COVID-19 patients who were in the recovery stage of the disease. Hospital care for the intervention group was transitioned to the patient’s home, including oxygen therapy, medication and remote monitoring. The control group received in-hospital care as usual. The primary endpoint was the number of hospital-free days during the 30 days following randomisation. Secondary endpoints included health care consumption during the follow-up period and mortality. Results: A total of 62 patients were randomised (31 control, 31 intervention). The mean difference in hospital-free days was 1.7 (26.7 control vs. 28.4 intervention, 95% CI of difference −0.5 to 4.2, *p* = 0.112). In the intervention group, the index hospital length of stay was 1.6 days shorter (95% CI −2.4 to −0.8, *p* < 0.001), but the total duration of care under hospital responsibility was 4.1 days longer (95% CI 0.5 to 7.7, *p* = 0.028). Conclusion: Remote hospital care for recovering COVID-19 patients is feasible. However, we could not demonstrate an increase in hospital-free days in the 30 days following randomisation. Optimising the intervention, timing, and identification of patients who will benefit most from remote hospital care could improve the impact of this intervention.

## 1. Introduction

Availability of hospital care is an ongoing challenge in the SARS-CoV-2 pandemic. Even with available vaccines, outbreaks might continue for years [1] and put pressure on hospital capacities. Particularly the need for oxygen therapy increases the length of hospitalisation and reduces the availability of beds. The addition of dexamethasone to supportive care improves the outcome but only shortens the length of stay by one day [2]. Hospitals therefore need to prepare for delivering long-term COVID-19 care while preserving regular care for other patients.

Admission for COVID-19 also has an impact on patient well-being. Family visits are limited to prevent viral spread, and patients are often transferred to referral hospitals in the case of capacity problems. Moreover, patients in contact isolation can present with depression, anxiety and anger [3]. The environment of a patient has been shown to influence a patient’s psychological well-being [4] and can be improved by a feeling of ‘being at home’ [4,5].

To ensure availability of hospital beds while improving patients’ well-being, hospital care can be offered at home. Hospital care at home has the potential to avoid admissions and reduce the length of in-hospital stay [6,7]. A pre-COVID-19 study reported a decrease of six in-hospital days per patient after implementing hospital care at home for acute patients [8]. This included daily visits by a trained nurse, which is labour intensive. Telemedicine is a less laborious alternative. Telemedicine offers healthcare-related services via electronic information and telecommunication technologies. These services can range from a single video consult to continuous remote monitoring, and these were successful in earlier studies [9]. During the pandemic, numerous hospitals have implemented telemedicine-based interventions to reduce the hospital stay for COVID-19 [10,11,12,13] or avoid hospital admission at all [12,14,15,16]. The first retrospective studies of reducing the length of stay by remote hospital care have shown promising results [10,11], but the added value has yet to be determined in a controlled setting.

In this trial, we assessed the effectiveness of remote hospital care for hospitalised, recovering COVID-19 patients. We hypothesize that transitioning hospital care to the home situation will result in more hospital-free days, without compromising patient safety.

## 2. Materials and Methods

### 2.1. Design and Setting

We conducted a non-blinded, randomised trial at the tertiary hospital in Utrecht, The Netherlands. Inclusion took place from 11 January to 7 May 2021. Since this centre was one of the primary referral centres for hospitals with capacity problems, 80% of COVID-19 patients were transfers and not necessarily in need of tertiary care. Patients were randomised 1:1 in either the intervention group or the control group. We used the block randomisation option in Castor with block sizes two, four and six (Castor Electronic Data Capture, Amsterdam, The Netherlands). The research ethics committee Utrecht approved the study (20/783). The study was registered in the Dutch Trial register (NL9081 Early@home).

### 2.2. Study Population

Eligible patients were identified by the treating physician at the ward and approached by a member of the research team, who double checked all in- and exclusion criteria. A patient could be included if he/she had confirmed COVID-19, was at least 18 years old, had a family member or other supportive caretaker at home, had a thermometer and smartphone, was able to use the mobile application (app) and pulse oximeter (possibly with help from the supportive caretaker), and was fluent enough in Dutch to understand the app. Patients were excluded if they suffered from dementia or other illnesses that limited the expected therapy compliance, if they needed more care than could be arranged at home, if the expected discharge destination was another care facility, e.g., a rehabilitation centre, or if discharge was already pending and hospital care was no longer needed. Patients were given 24 h to consider participation. After signing informed consent, the General Practitioner (GP) was contacted to check whether the home situation was safe enough for the intervention. Randomisation was performed when the patient met all discharge criteria: a maximum of 3 L/min oxygen therapy with no increase during the last 24 h, no intravenous medication that cannot be replaced with a non-intravenous alternative, no planned diagnostic tests that needed to be performed in the hospital or could lead to in-hospital treatment, permission by the treating physician. 

### 2.3. Care as Usual

The control group received care as usual. Vital signs were checked three times daily, or more in the case of clinical instability. As a rule, patients were discharged 24 h after oxygen therapy was ceased, but the treating physician could make a substantiated decision to differ from this rule. A standard dose of dalteparin to prevent thrombosis due to immobilisation was given until discharge, and dexamethasone was given until the end of oxygen therapy, with a maximum of ten days. At the start of this trial, oxygen therapy, dexamethasone and dalteparin administration were considered in-hospital treatments in The Netherlands.

### 2.4. Intervention

For the intervention group, remote hospital care was organized as soon as possible after randomisation. The hospital care provided at home, instead of in the hospital, included oxygen therapy and prescriptions for necessary medication. Visits by hospital staff were replaced with remote monitoring and telephone contact (see below). Dexamethasone and prophylactic dalteparin administration was similar to the control group, with an exception for active patients, who were no longer required to receive thromboprophylaxis. Patients received a medically certified pulse oximeter (iHealth^®^ Air, Andon Health Co. Ltd., Tianjin, China) and an account for the app (Luscii Healthtech BV, Amsterdam, The Netherlands). They received information from the Medical Control Centre (MCC), a group of trained medical students who performed the monitoring. After inclusion in the intervention group, a member of the MCC would visit the patient to inform him/her on how to use the app and to provide the patient with a manual and a personal treatment plan, including personalised thresholds for oxygen saturation (Appendix A).

While at home with remote hospital care, patients filled out a questionnaire in the app three times daily at predetermined moments. The questionnaire consisted of scores for coughing, shortness of breath and general well-being, temperature and oxygen saturation (Appendix A). The MCC checked these questionnaires between 09:00 and 16:00, including the questionnaires from the prior evening. The patient was called immediately in the case of irregularities that required rapid action. Between 11:00 and 12:00, the MCC called all patients by telephone for a daily check-up and to communicate any changes in treatment, such as titration of oxygen therapy, changes in medication, or readmission in case of severe deterioration. A consultant in internal medicine with COVID-19 expertise supervised the MCC and was involved in all treatment decisions. If oxygen therapy was ceased for 48 h and the patients’ condition was stable, the patient was discharged from hospital responsibility.

### 2.5. Follow Up

At 30 days after randomisation, the patient was contacted for follow-up on the number of hospital-free days. If a patient could not be reached, the GP provided the missing information.

### 2.6. Endpoints and Data Collection

The primary endpoint was the number of hospital-free days [17]. This patient-centred endpoint includes both the index hospital length of stay and the impact of readmissions [18,19,20]. The number of hospital-free days is defined as the number of days spent alive at home in the 30 days following randomisation. For every day the patient was readmitted or was not alive after randomisation, a day was subtracted. Since emergency department (ED) visits and other unplanned hospital visits often result in a day spent in the hospital too, these were added to the score [17]. Secondary endpoints were length of hospital stay, length of hospital stay and oxygen therapy following randomisation, duration of care under hospital responsibility, number of readmissions, ED visits, other unplanned hospital visits and GP contacts, and mortality. Care under hospital responsibility was defined as the total number of days a patient was under the responsibility of the hospital, either in the hospital or at home with remote hospital care. If the end of treatment was not reached by the time of follow-up, the follow-up date was registered as end-date. To describe the cohort, admission information and patient characteristics were documented, including the Charlson Comorbidity Index [21] and Clinical Frailty Scale score [22]. 

### 2.7. Sample Size 

Since the primary outcome was expected to be left skewed (most patients will reach a high number of hospital-free days), the sample size was determined using 1000 simulations based on the Mann–Whitney U test. We based the power on a one-day difference, the minimal clinically relevant difference. Since we assumed some variety in the effect of the intervention, we added a distribution of hospital stay following randomisation in both groups (Appendix A). Previous studies showed a readmission rate of 2.2–18% (mean 9%) [23,24,25,26]. Since our study did not include patients who were discharged to a nursing home or rehabilitation centre, which is one of the primary risk factors for readmission [23], we assumed a lower readmission rate of 5%. For the intervention group, we assumed a readmission rate of 9% [27]. We assumed a mean readmission length of 5 days. Mortality was not accounted for since the expected mortality in this population of recovering patients was low. With these assumptions, 80% power and a two-sided alpha of 0.05, a sample of 62 patients was needed. 

### 2.8. Statistical Analysis

Primary analysis was based on intention-to-treat. Given the skewed distribution, we used the bootstrap *t*-test with 10,000 iterations to calculate the mean difference in number of hospital-free days with a 95% confidence interval [28]. The same approach was used to compare the secondary endpoints for duration of care. The Wald statistic was used to test the difference in the number of GP contacts. Since the study was highly underpowered to find differences in the remaining secondary endpoints, these results were presented descriptively. A per-protocol sensitivity analysis was performed since not all patients were treated as intended. Because an increasing number of patients in the control group were discharged with oxygen therapy at home, which is not a complete intervention but also differs from usual care, we looked at the characteristics of this group separately. A two-sided alpha of 0.05 or lower was considered statistically significant. RStudio version 4.0.3 was used for all analyses.

## 3. Results

Of the 226 patients selected by the ward physicians, 48 did not meet the inclusion criteria at the second evaluation. Another 27 patients did not feel safe towards the idea of remote hospital care, and 41 patients did not consent for other or unknown reasons. Three patients agreed to participate but dropped out before randomisation (Figure 1). Ultimately, 62 patients were included. One patient changed his mind after learning he was randomised to the intervention group and was discharged home using the usual care route. Baseline characteristics of participating patients are shown in Table 1. The included patients were relatively young and had few comorbidities.

As shown in Table 2, the mean difference in hospital-free days between the two groups was 1.7 days (95% CI −0.5 to 4.2, *p* = 0.112) but was not statistically significant. This result was not altered by the per-protocol analysis (difference of 1.7 (95% CI −0.6 to 4.2, *p* = 0.126)). The distribution of hospital-free days is shown in Figure 2. We observed several differences in secondary endpoints. Patients in the intervention group had a shorter index admission stay following randomisation and were able to go home 1.6 days earlier than patients in the control group. In contrast, they received longer oxygen therapy and care under hospital responsibility than the control group (Table 2). We found no difference in the number of days in the hospital or death after index admission. Patients in the control group made 2.4 times more visits to the GP, mostly for COVID-19 (Table 2). Twenty-five patients in the intervention group recorded one or more values below the threshold for oxygen saturation while at home, and eighteen patients recorded an oxygen saturation of 92% or lower at some point during remote monitoring.

Five patients in the control group received oxygen therapy at home. Baseline characteristics were comparable with the rest of the cohort. The patients were discharged 0–3 days after randomisation; they reached a number of hospital-free days of 27, 27, 29, 29 and 30 days. All of these patients visited their GP and none of them were readmitted or presented at the ED.

## 4. Discussion

In this study, we assessed the effectiveness of remote hospital care for recovering COVID-19 patients. Although remote hospital care using telemedicine was feasible, we were unable to show a significant increase in hospital-free days. The initial reduction in index length of stay did not result in a significant difference in hospital-free days after 30 days, probably due to a different distribution in hospital-free days than expected.

### 4.1. Comparison with Previous Research

The reduction in length of index admission of 1.6 days was smaller than reported previously [10,11]. The difference in effect size between studies is most likely caused by the difference in duration of oxygen therapy inside and outside the hospital. Previous studies counted every day of oxygen therapy at home as a reduction in hospital length of stay. Our study shows that oxygen therapy was tapered down more slowly at home than at the hospital. If all these days are counted as a reduction in hospital stay, the effect size could be overestimated. Secondly, we found that it was difficult to predict the duration of oxygen therapy, making it hard to determine which patients would benefit from remote hospital care. If future research can identify those patients who will benefit most, the impact of the intervention might be higher. Furthermore, contamination of the control group in our study has occurred. As the study progressed, clinicians recognized that earlier discharge was possible and applied this to the control group too, sometimes even with oxygen therapy at home. Disappointment of patients in the control group might have amplified this since they often asked if they could leave the hospital early anyway after learning the outcome of randomisation. This is highlighted by our observation that oxygen therapy in the control group was tapered down in one day, even though mean oxygen administration at randomisation was 2 L/min.

Although the index admission was shorter, the total length of care under hospital responsibility was longer in the intervention group. This was partly due to protocol. Patients in the control group were discharged as soon as the treating physician saw fit, but patients in the intervention group were only discharged two days after oxygen therapy was ceased. However, a prolonged length of hospital responsibility is not necessarily a negative outcome. The number of GP visits for COVID-19 was more than two times higher in the control group, indicating a need for patients to be in contact with a physician following hospital admission. While tapering oxygen therapy took longer at home, patients were more in charge of their own therapy. Given the regular low values for oxygen saturation in the intervention group, COVID-19 patients in general might benefit from a more gradual tapering of oxygen therapy, something that cannot be accomplished in a pressured hospital environment. Remote hospital care could be an intermediate step between the hospital and home, giving patients a little more time to recover and regain confidence in their bodies. Earlier studies report that patients appreciate the intervention, feel safe at home, and would recommend it to others [11,27]. 

### 4.2. Strengths and Limitations

Several retrospective studies studied the effect of remote hospital care for recovering COVID-19 patients, but this is the first study to compare the intervention in a controlled randomised setting. Although this was a single centre study in a tertiary hospital, the majority of patients were allocated from hospitals across the country, creating a varied population. The fluctuating case-loads during the pandemic interfered with the study. Variable hospital capacity might have influenced decision making by physicians. As the pandemic progressed, clinicians gained confidence in treating COVID-19 patients, resulting in faster tapering of oxygen therapy and more patients being discharged with oxygen therapy. Unfortunately, blinding was not possible, which resulted in contamination. The predetermined variation of the expected effect size (Appendix A) will only have partially corrected for this. Although this randomised trial was designed with practicality in mind, the artificial circumstances might have led to a delay in randomisation, for example, due to the mandatory 24 h consideration period. Furthermore, some patients were willing to participate in the intervention but not in a study setting, leading to selection bias.

Even though we found no evidence that the intervention was unsafe, this finding should be interpreted with caution. Major differences in safety would have influenced the primary endpoint and would have become apparent even in a small population. However, our study was not powered to find definite differences in mortality or readmission rate, for which much larger trials would be needed.

### 4.3. Implications for Future Practice

The basic elements of this intervention (oxygen therapy and medication at home combined with remote monitoring) can be used in a variety of countries and settings. All over the world, remote monitoring has already been implemented for this purpose [29,30,31,32], and several have added home treatment using comparable inclusion criteria [12,24,31], showing the high level of feasibility. Nevertheless, the intervention is only feasible in a limited group of patients. In all studies, the patients included were relatively young, healthy, and capable of adequate communication with hospital staff [10,11,12]. Only 8% of all admitted patients in the study of van Herwerden et al. and 10% of the admitted patients in our study ultimately received remote care. Several steps could be taken to expand usability. The intervention could be made suitable for more patients, for example elderly, by making the app more accessible or less vital. Secondly, if we could predict who will benefit most of this intervention, the success rate could be improved. Lastly, the timing of the intervention could be optimized. Since a considerable number of eligible patients were discharged before inclusion could take place, the inclusion criteria might have been too strict. Nonetheless, the effect of these alterations on the safety of the intervention should be carefully monitored.

## 5. Conclusions

In conclusion, remote hospital care for recovering COVID-19 patients was feasible, but we were unable to show an increase in the number of hospital-free days 30 days after randomisation. By optimising the intervention, the timing of the intervention, and identification of those patients who will benefit most, the impact might be considerably improved.

## Figures and Tables

**Figure 1 jcm-10-05940-f001:**
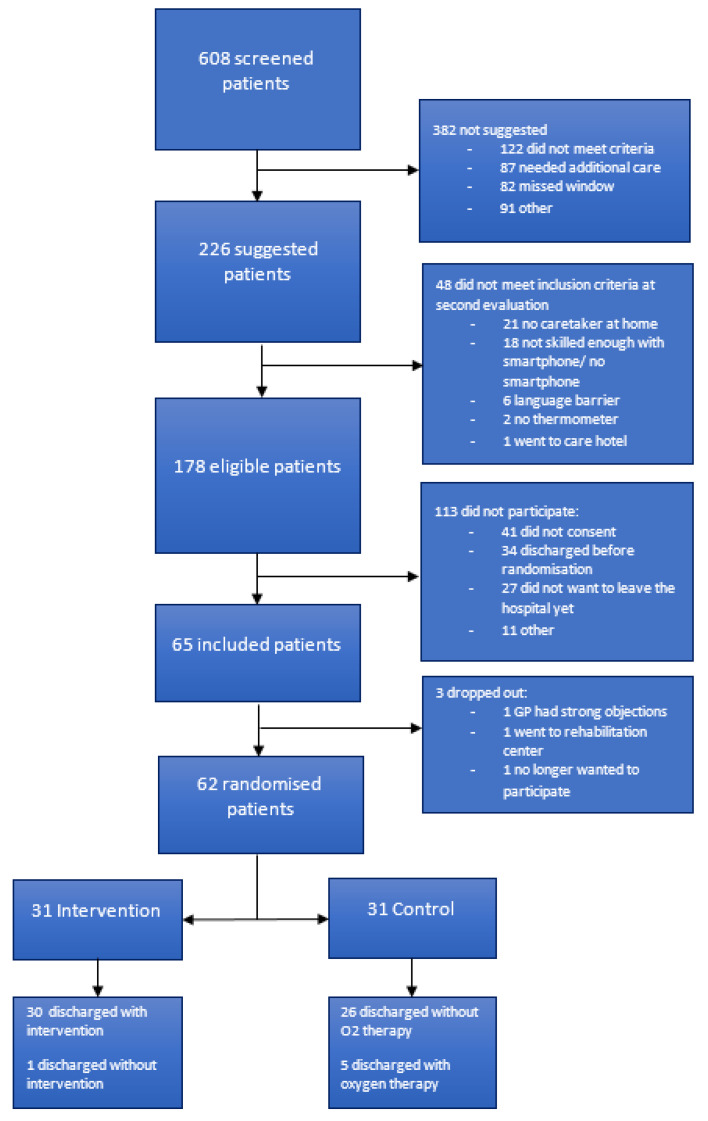
Flowchart of study inclusion. GP—general practitioner.

**Figure 2 jcm-10-05940-f002:**
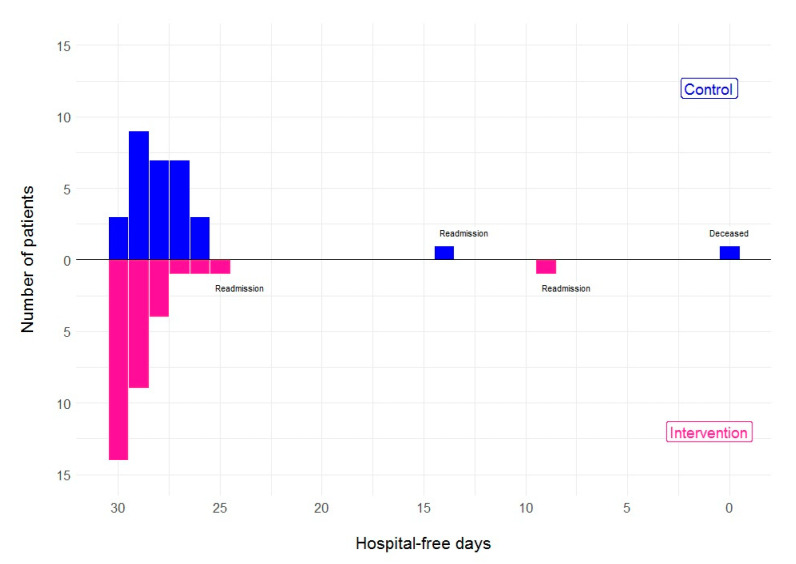
Distribution of number of hospital-free days 30 days after randomisation.

**Table 1 jcm-10-05940-t001:** Cohort description.

	Control (*n* = 31)	Intervention (*n* = 31)
Patient		
Age (mean, sd)	55.4 (13.2)	55.1 (7.5)
Female (%)	13 (41.9%)	14 (45.1%)
CFS * (mean, sd)	2.1 (1.3)	2 (0.6)
Active smoker (%)	0 (0%)	1 (3.2%)
Hypertension (%)	5 (16.1%)	6 (19.4%)
Cardiovascular disease (%)	3 (9.7%)	2 (6.5%)
CCI ** (median, IQR)	2 (0–3)	1 (1–2)
Index admission		
Transferred from a different hospital (%)	26 (83.9%)	28 (90.3%)
Admitted to ICU (%)	3 (9.7%)	4 (12.9%)
Length of hospital admission before randomisation (median, IQR)	6 (4.5–9)	6 (4–8.5)
Pulmonary embolism (%)	3 (9.7%)	2 (6.5%)
Bacterial superinfection (%)	3 (9.7%)	2 (6.5%)
Other (%)	1 (3.2%)	3 (10%)
Dexamethasone or prednisone treatment (%)	31 (100%)	31 (100%)
Highest delivered FiO2 at ward (median, IQR)	0.44 (0.36–0.6)	0.4 (0.36–0.6)
Oxygen therapy at randomisation (L/min) (mean, sd)	2.0 (1.0)	2.1 (0.9)
Discharged from hospital care with oxygen therapy (%)	5 (16.1%)	1 (3.2%)

* CFS—clinical frailty scale, ** CCI—Charlson comorbidity index. One patient in the intervention group was discharged from hospital care while on oxygen therapy; she was handed over to the outpatient clinic of her own pulmonologist.

**Table 2 jcm-10-05940-t002:** Comparison of main outcomes.

	Control (*n* = 31)	Intervention (*n* = 31)	Difference (95% CI)	*p*-Value
Hospital-free days in 30 days following randomisation (mean, sd)	26.7 (5.7)	28.4 (3.8)	1.7 (−0.5 to 4.2)	0.112 *
Index hospital length of stay (mean, sd)	10.0 (7.0)	7.3 (4.3)	−2.7 (−5.7 to 0.0)	0.045 *
Duration of index hospital stay after randomisation (mean, sd)	2.3 (2.3)	0.7 (0.9)	−1.6 (−2.4 to –0.8)	<0.001 *
Number of days in hospital or dead following index hospital stay (mean, sd)	1.0 (3.7)	0.9 (3.7)	−0.1 (−2.1 to 1.8)	0.906 *
Duration of hospital responsibility (hospital stay + hospital care at home) (mean, sd)	10.0 (7.0)	14.1 (7.6)	4.1 (0.5 to 7.7)	0.028 *
Days of oxygen therapy following randomisation (mean, sd)	3.4 (7.5)	6.7 (7.5)	3.3 (−0.5 to 6.8)	0.101 *
ED visits (*N*, %)	1 (3.2%)	3 (9.7%)	–	–
COVID-19	1	3		
Other unplanned hospital visits (*N*, %)	2 (6.5%)	2 (6.5%)	–	–
- For COVID-19	2	2	–	–
Readmission (*N*, %)	1 (3.2%)	2 (6.5%)	–	–
- For COVID-19	1	2		
GP visits (*N*, %)	20 (64.5%)	12 (38.7%)	–	0.035 †
- For COVID-19	19	8		
Telephone contact with GP by patient (%)	22 (71.0%)	25 (80.6%)	–	0.371 †
Mortality (%)	1 (3.2%)	0 (0%)	–	–

* Bootstrap *t*-test with 10,000 iterations; † risk ratios with Wald statistic; sd—standard deviation; ED—emergency department; GP—general practitioner.

## Data Availability

The data presented in this study are available on request from the corresponding author. The data are not publicly available due to the privacy sensitivity of the data.

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
