# Peer review of "Remote Hospital Care for Recovering COVID-19 Patients Using Telemedicine: A Randomised Controlled Trial"

_jcm, 2021, doi:10.3390/jcm10245940_

Round 1

Reviewer 1 Report

1- The authors should add the most recent works in the reference list.

2- The proposed method need to be explained in readable and simplicity .

3- the authors can use machine learning tools as deep learning in the case study.

Author Response

Dear reviewer,

Thank you very much for your thorough review. We have taken note of your recommendation that the manuscript needs extensive editing of English language and style. The manuscript has been edited by two qualified colleagues, whose suggestions to improve the manuscript were all accepted. Regarding your additional concerns, we have addressed them as follows: 

  1. We have done some extensive searching and have added several, more current, references to the manuscript (references 13, 15, 16).
  2. We have explained some aspects of the inclusion criteria as suggested by reviewer 3, and we have rephrased several sentences. We hope that, with these adjustments, you’ll find the manuscript sufficiently readable.
  3. Thank you for this suggestion, we have indeed considered the use of machine learning tools in the case study. Machine learning is a very powerful tool, especially in diagnosing and predicting diseases and outcomes. However, we do not feel like machine learning has a place in the telemedicine intervention described in our study. Even though we mention that the intervention can be improved by the right timing and identification of patients, this should not necessarily be done by machine learning. Conventional (regression) techniques might be sufficient to predict the right timing and patient population, and are generally easier to understand by the users of clinical decision models. We therefore decided not to refer to machine learning tools in our manuscript.

Reviewer 2 Report

My congratulations to the authors for the quality of the manuscript.

The manuscript is clear, relevant for the field and presented in a well-structured manner and the cited references are current.
The manuscript is scientifically sound and the experimental design are appropriate to test the hypothesis.
The major limitation of the study has to do with the nature of the intervention, which, in addition to being complex, cannot be hidden to the participants. The authors make an excellent reflection on this aspect.
Despite not showing outcome measures with evident differences between groups. demonstrates that the remote hospital care for recovering covid-19 patients is feasible.

Author Response

Dear reviewer,

Thank you for the kind words. The intervention indeed has some intrinsic limitations, and we are glad to hear that you feel we reflected upon these limitations adequately. We are very grateful for your review of our manuscript.

Reviewer 3 Report

In the paper by van Goor and colleagues, the authors addressed a relevant issue describing the strategy they used to build up a telemedicine service for patients affected by COVID-19. Despite the topic is of interest, the enthusiasm of the reviewer is reduced by some limitations (unblinded single centre study, COVID-19 pandemic fluctuation) and minor concerns:

Line 46: please correct the [4] reference;
Line 58: please add this reference 10.1136/annrheumdis-2020-218022;
Line 79: please defined what do you mean as “patient… was expected to be discharged home”

Author Response

Thank you for your thorough review and the specific pointers. We have addressed them as follows:

  • Line 46: we have corrected reference [4].
  • Line 58: thank you for suggesting this additional reference. After reading the letter, we have chosen not to use this particular example, since it concerns telemedicine for patients with rheumatic disease more than it concerns telemedicine for patients recovering from covid-19. However, we did add additional and up-to-date references to the manuscript to further improve the theoretical foundation.
  • Line 79: we assume this comment is about line 86 (instead of 79). By ‘expected to be discharged home’, we meant that the expected discharge destination of a patient was his/her home, and not a care facility such as a rehabilitation centre, hospice, or nursing home. For clarification, we have removed this from the inclusion criteria and we have added a more elaborate explanation to the exclusion criteria in line 91/92.